# Spatial Insights into Drought Severity: Multi-Index Assessment in Małopolska, Poland, via Satellite Observations

Jakub Staszel [1], Michał Lupa [1,*], Katarzyna Adamek [1], Michał Wilkosz [1], Adriana Marcinkowska-Ochtyra [2] and Adrian Ochtyra [2]

1   Department of Geoinformatics and Applied Computer Science, Faculty of Geology, Geophysics and Environmental Protection, AGH University of Krakow, 30-059 Krakow, Poland; jstaszel@agh.edu.pl (J.S.); kadamek@agh.edu.pl (K.A.); mwilkosz@agh.edu.pl (M.W.)
2   Department of Geoinformatics, Cartography and Remote Sensing, Faculty of Geography and Regional Studies, University of Warsaw, 00-927 Warsaw, Poland; adriana.marcinkowska@uw.edu.pl (A.M.-O.); adrian.ochtyra@uw.edu.pl (A.O.)
*   Correspondence: mlupa@agh.edu.pl

**Abstract:** This study focuses on the assessment of drought severity, employing a comparative analysis between the normalized multi-band drought index (NMDI; calculated using Sentinel-2 imagery) and the combined drought indicator (CDI). The research aims to pinpoint the most accurate and reliable remote sensing techniques, which reflect ground-based measurements, thereby enhancing the precision of drought monitoring systems. By investigating the specific area of Małopolska, the study not only contributes to the global knowledge base on drought assessment methodologies but also addresses the regional needs for improved drought management practices. Through an exhaustive analysis correlating satellite-derived indices with in situ meteorological data, this research elucidates the potential of integrating NMDI and CDI for a comprehensive understanding of drought dynamics in Małopolska. In particular, the indices depict different dynamics of drought levels, as well as the location of regions more prone to its occurrence. The findings are poised to advance drought monitoring capabilities, offering significant insights for agricultural sustainability and water resource management in the region.

**Keywords:** drought; CDI; NMDI; remote sensing; combined drought indicator; normalized multi-band drought index; Małopolska; Lesser Poland

## 1. Introduction

As the global climate continues to change, the frequency and severity of droughts are increasing, posing significant challenges to water resource management, agriculture and ecosystems worldwide. Climate change has significant implications for drought patterns, with various studies highlighting the potential exacerbation of water scarcity. Schewe [1] emphasizes that climate change is likely to considerably worsen regional and global water scarcity. This is supported by Apostolaki [2], who demonstrates that climate change, associated with increased temperature and decreased precipitation, imposes high stress on water resources, leading to increased water scarcity and drought. Furthermore, Graham [3] asserts that future changes in climate and socio-economic systems will drive both the availability and use of water resources, resulting in evolutions in scarcity.

The impact of climate change on drought severity is also a key concern. Wang [4] highlights that the intensity, frequency and duration of droughts are expected to increase due to climate change, particularly for agricultural and hydrological droughts. Additionally, Verdon-Kidd and Kiem [5] express serious concern about how anthropogenic climate change may exacerbate drought risk in the future. Moreover, Brownlee [6] projects an increasing trend in areas affected by drought due to global climate change.

The influence of climate change on water scarcity and drought is not limited to specific regions. Butts [7] presents a methodology for assessing climate change impacts and adaptation potential for floods, droughts and water scarcity within the Nile Basin, emphasizing the regional approach to climate adaptation. Furthermore, Abedin [8] highlights the potential health consequences of climate change, particularly malnutrition and water scarcity, affecting a large number of people in southwestern coastal Bangladesh.

Drought itself is a multi-faceted phenomenon, which is difficult to define. Challenges arise primarily due to differences in hydrological and meteorological variables, socio-economic factors and the diverse nature of water demand in various regions of the world [9]. Emerging definitions can be categorized as conceptual and operational. Conceptual definitions use general terms and cannot be applied to the assessment of current drought; for example, "drought is a longer period without rain". On the other hand, operational definitions attempt to define aspects such as the onset, intensity and end of drought episodes. Often, through the evaluation of impacts, such as the influence of rainfall, soil moisture and evapotranspiration on crop quality, such studies can be conducted systematically throughout the growing period (almost in real time) [10]. The most recognized classification, facilitating communication, management and response, distinguishes four types of droughts: meteorological, agricultural, hydrological and socio-economic [10].

The Intergovernmental Panel on Climate Change (IPCC), in its sixth report (Climate Change 2023: Synthesis Report) [11], highlights that with advancing climate change, the occurrence of extreme weather events intensifies (such as heavy rainfall, meteorological and agricultural droughts), which can be observed in every region of the world. The indisputable cause of these climate changes is attributed to humans, whose activities have led to widespread and abrupt alterations in the atmosphere, oceans, cryosphere and biosphere.

Human activities have significantly impacted natural hydrological processes, particularly in the context of drought. The alteration in soil moisture, hydrological cycle and groundwater recharge due to human-induced changes in land and water management has led to the reframing of drought definitions and understanding [12,13]. Anthropogenic drought is now recognized as a compound, multi-dimensional phenomenon influenced by natural water variability, climate change, human decisions and altered micro-climate conditions [14]. Human activities, such as land cover change, reservoir regulation and agricultural irrigation, have been identified as significant contributors to hydrological drought in various regions [15,16]. Additionally, the construction of dams and reservoirs in arid regions has raised concerns about their impacts on hydrological droughts [17]. Furthermore, anthropogenic warming has been found to substantially increase the likelihood of extreme drought events, such as those observed in California [18]. The influence of human activities on water availability has been highlighted, emphasizing the need to separate meteorological variability from anthropogenic impacts using satellite observations [19]. Moreover, the occurrence of unprecedented drought events, such as the 2018–2019 Central European drought, has been linked to global warming and human-induced climate change [20].

## 1.1. Monitoring Drought with Remote Sensing

Changes and progress in monitoring drought have been the subject of extensive research and development in recent years. Efforts have been made to enhance drought monitoring, forecasting and early warning systems, as well as to improve research on the combined effects of anthropogenic activities and changes in climate systems [21]. Historically, assessments relied solely on in situ measurements, which did not provide sufficient spatial–temporal accuracy. Since 1972, with the launch of the Landsat mission by the National Aeronautics and Space Administration (NASA), the approach began to change dramatically, as these events revolutionized the field of drought monitoring. This has been particularly true in recent years, with the addition of European Space Agency (ESA) Sentinel satellites to earth-monitoring missions. There has been a significant increase

in interest within the scientific community in drought monitoring techniques using remote sensing. The number of articles on this topic in the *Remote Sensing of Environment* journal increased from 5 in 1982 to over 70 in 2017 [22].

The use of remote sensing technology has significantly advanced drought monitoring in Europe. Remote sensing has proven to be a powerful tool for assessing the temporal and spatial aspects of drought events [23]. It has revolutionized the field by allowing observations and monitoring of key drought-related variables over larger temporal and spatial scales than was previously possible using conventional methods [22]. The remotely sensed global terrestrial drought severity index (DSI) enhances the capabilities for near-real-time drought monitoring to assist decision makers in regional drought assessment and mitigation efforts [24]. Additionally, the combined drought indicator (CDI) has been successfully applied within the European Drought Observatory (EDO) as part of a near-real-time monitoring with decadal updates [25]. Furthermore, remote sensing data play an important role in drought monitoring, especially in studying the spatiotemporal dynamics of drought, due to their multi-temporal sampling and high-resolution spatial coverage [26]. In the context of agricultural drought monitoring, remote sensing has been instrumental in providing a comprehensive assessment of regional drought by considering factors such as soil water stress, vegetation growth status and meteorological precipitation [27]. It has also been used to develop a series of remote sensing indicators for crop drought monitoring, such as the temperature vegetation drought index (TVDI) [28]. Moreover, the use of remote sensing-based drought indices has been evaluated using cloud-based computing platforms, demonstrating the versatility and adaptability of remote sensing technologies in drought monitoring [29]. The normalized multi-band drought index (NMDI) has emerged as a valuable tool for monitoring soil and vegetation moisture using remote sensing data. NMDI integrates information from multiple near-infrared and short-wave infrared bands, making it more sensitive to drought severity and capable of estimating soil moisture and vegetation conditions [30]. This index has been successfully applied in the evaluation of vegetation stress by soil water [31] and monitoring forest fires due to its accuracy in assessing drought severity [32]. Additionally, NMDI has been developed based on the fact that short-wave infrared (SWIR) is more responsive to soil and vegetation moisture, further improving the index's sensitivity for drought severity monitoring [30]. NMDI has also been used to identify dry soil or bare land, making it a versatile indicator for various environmental assessments [33]. In the case of satellite-based monitoring of atmospheric precipitation and its use in drought assessment, indices such as the precipitation condition index (PCI) [34] and the standardized precipitation index (SPI) [35] are utilized. Evaluation of evapotranspiration is also possible through algorithms such as the global land evaporation Amsterdam model (GLEAM) [36]. The integration of remote sensing, modeling and monitoring data has been emphasized as crucial for evaluating droughts and establishing a comprehensive understanding of the linkages between meteorological and hydrological droughts for future management [37]. Additionally, the primary goal of using optical and thermal remote sensing in the monitoring, assessment and prediction of agricultural drought has been highlighted, indicating the diverse applications of remote sensing in addressing drought challenges [38].

### 1.2. Area of Interest

Climate change impacts in Central Europe have been the subject of extensive research. Studies have shown that the region is experiencing significant changes in climate variables, with observed warming signals in temperature [39]. The impacts of climate change are multi-faceted, affecting various aspects of the environment, including vegetation, hydrology and ecosystems. For instance, there is evidence of increased forest growth, particularly in Northern Europe, as a result of climate change [40]. Additionally, changes in the North Atlantic thermohaline circulation have been found to significantly impact weather conditions and climate elements in Poland [41].

Furthermore, climate change is projected to have substantial effects on water resources in Central and Eastern Europe, with warmer climates expected to lead to shifts in the area and location of agroclimatic zones [42,43]. This is particularly relevant, given the potential for more severe and impactful drought events in the future [44]. Moreover, the impacts of climate change extend to ecological networks, with studies indicating that these networks are more sensitive to plant than to animal extinction under climate change [45]. The observed changes in climate variables have also led to concerns about extreme weather events, such as heatwaves, droughts and floods. For instance, the summer climate in Europe has shown a drying trend, accompanied by devastating drought and flood events [46]. Additionally, there is evidence of a record dry summer in 2015, challenging precipitation projections in Central Europe [47]. These extreme events have implications for various sectors, including agriculture, biodiversity and public health [48,49].

The Małopolska region in Poland presents a compelling case for studying the impacts of climate change. This region is characterized by diverse ecosystems, including the Carpathian Mountains, which are particularly sensitive to climate variations [50]. The region's vulnerability to climate change is further underscored by the potential impacts on agriculture, as evidenced by the urgency to address agricultural adaptation in the face of increasing climate impacts [51]. Additionally, the FORESEE database was developed specifically to support research and adaptation to climate change in Central and Eastern Europe, emphasizing the inadequate knowledge of possible climate change effects in this region [52].

Furthermore, the Małopolska region's agricultural sector is of significant importance, and the impacts of climate change on crop productivity are a crucial area of study [53]. An evaluation of the quality of the NDVI3g (third generation of the normalized difference vegetation index developed by Global Inventory Modeling and Mapping Studies) dataset in Central Europe also highlights the relevance of studying the region's vegetation activity and greenness in the context of climate change impacts [54].

The diversity of the landscape, characteristic of Małopolska (Figure 1), is largely due to the terrain's topography, as well as hydrological and soil conditions. The mentioned topography promotes the runoff of precipitation waters, making this region susceptible to climatic changes through the potential occurrence of hydrological drought phenomena. It is worth noting that the Małopolska region is characterized by the highest variability in water flows in Poland. This variability, combined with diversified land morphology, makes it difficult to develop a uniform strategy to combat extreme weather phenomena for this region.

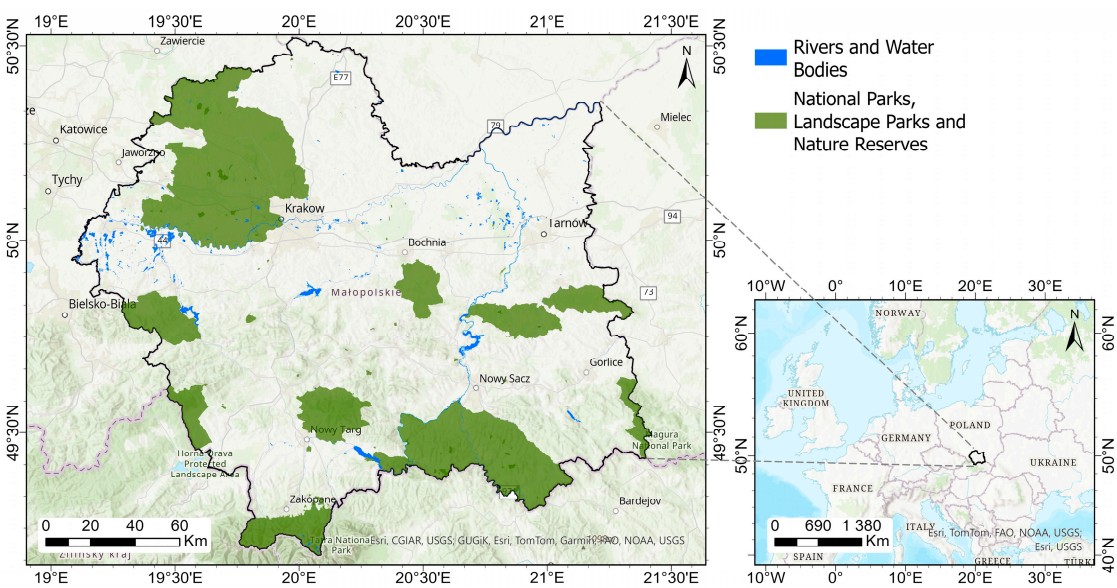

**Figure 1.** Location of Małopolska region.

Małopolska is also a diverse region in terms of land use (Figure 2). In the north, there are areas extensively utilized for agriculture. Below, urbanized areas are located, with the largest ones being the cities of Kraków and Tarnów. Smaller urbanized areas can be found in the whole area. Further to the south are mountains covered with vegetation. In the valleys, urbanized areas can be observed, but they are also used for agriculture. Most of the mountains in Małopolska are covered with vegetation, except for the Tatra Mountains located in the far south.

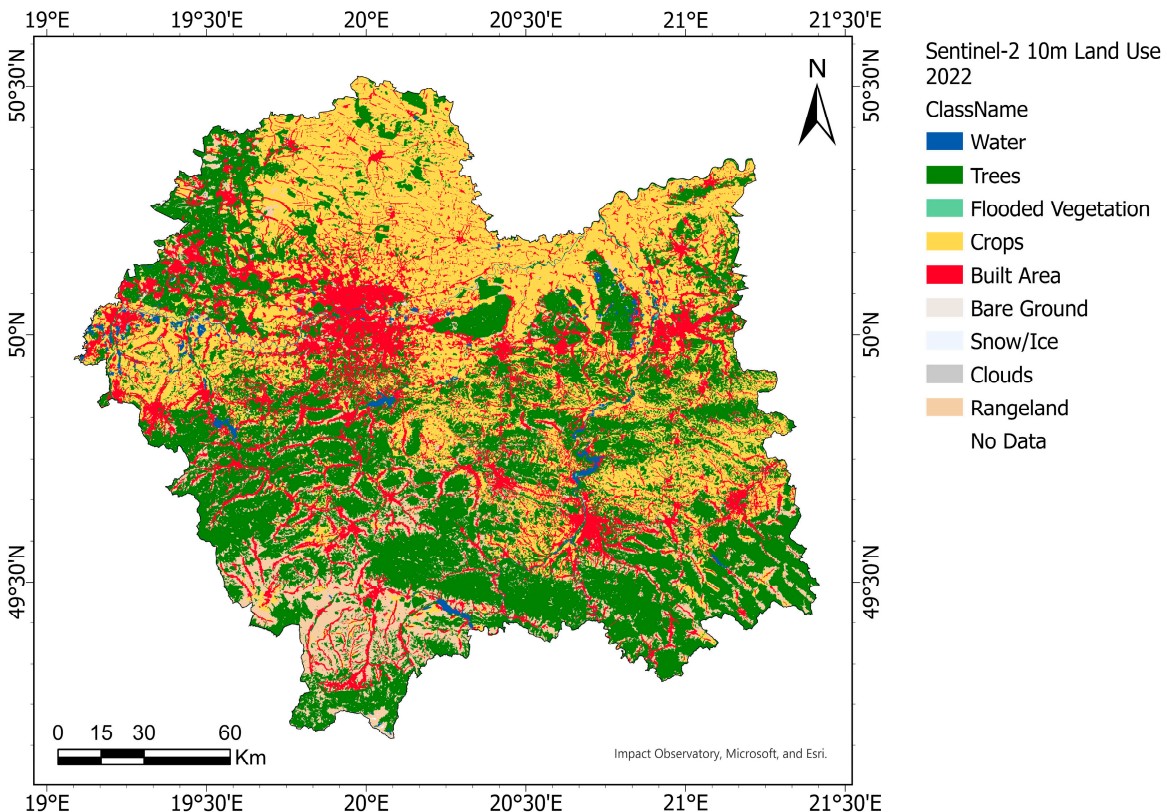

**Figure 2.** Land use/land cover for Małopolska region using Sentinel-2 for the year 2022; accessed through ArcGIS Living Atlas.

As the report [55] shows, the trend values of average and maximum daily temperature in the Małopolska region are alarming. The annual average maximum temperature in a significant part of the region shows strong positive trends, reaching around 4.0 °C. Referring to the Intergovernmental Panel on Climate Change report (IPCC; 2001, 2013) [56,57], climate changes are often understood precisely through changes in extreme temperatures. Małopolska is a region where 60% of the area is legally protected by nature conservation (6 national parks, including the largest in Poland—Tatra, 11 landscape parks, 85 nature reserves, NATURA 2000 areas and others). Therefore, if the tendency of temperature increases continues in the coming decades, the ecological diversity of this region may be permanently lost.

### 1.3. Research Focus

The research objective of this study is to investigate various drought assessment methods and their application for the Małopolska region. In situ measurements, despite their highest reliability, often lack sufficient temporal and spatial coverage for a comprehensive assessment of the environmental conditions. Remote sensing technologies seem to provide solutions, raising the question of whether they can extend the application of traditional methods. This paper examines remote sensing methods for drought assessment, compares them and attempts to correlate these data with in situ measurements. The study investi-

gates whether the results from both methods overlap, meaning they similarly represent the temporal variability in drought levels in Małopolska, and whether they identify regions more prone to drought in a similar manner.

Despite the widespread use of the combined drought indicator, which relies on historical data and employs a more matured methodology utilizing multiple data sources, its spatial resolution is insufficient—5 km. Therefore, the following question arises: is it possible to utilize methods with higher spatial resolution? To address this question, the interpretation of the normalized multi-band drought index was chosen because it directly pertains to drought. Moreover, with the use of Sentinel-2 data, it can provide information with a spatial resolution of 20 m and, under favorable conditions, every 5 days. These features seem ideal for applying NMDI in the creation of early warning systems for drought occurrences.

The research interests of this study extend beyond the assessment methods for drought alone, encompassing the overall condition of the Małopolska region. The focus is not only on the general occurrence of drought-prone areas but also on the spatial distribution of regions with an elevated risk of drought within Małopolska.

Therefore, the main research hypothesis of this study is as follows: the utilization of high spatial resolution satellite imagery is an excellent source of information regarding drought, which yields similar results to the application of in situ measurements or more matured remote sensing methods.

## 2. Materials and Methods

An overview of the materials and methods used in this study is presented in the diagram below (Figure 3). Data preparation consists of acquisition with some initial pre-processing. The final datasets are generated using interpreted and down-sampled NMDI, contributing with CDI to the analysis part, consisting of comparative maps and statistical analysis. In-depth descriptions of the different steps can be found in the following paragraphs.

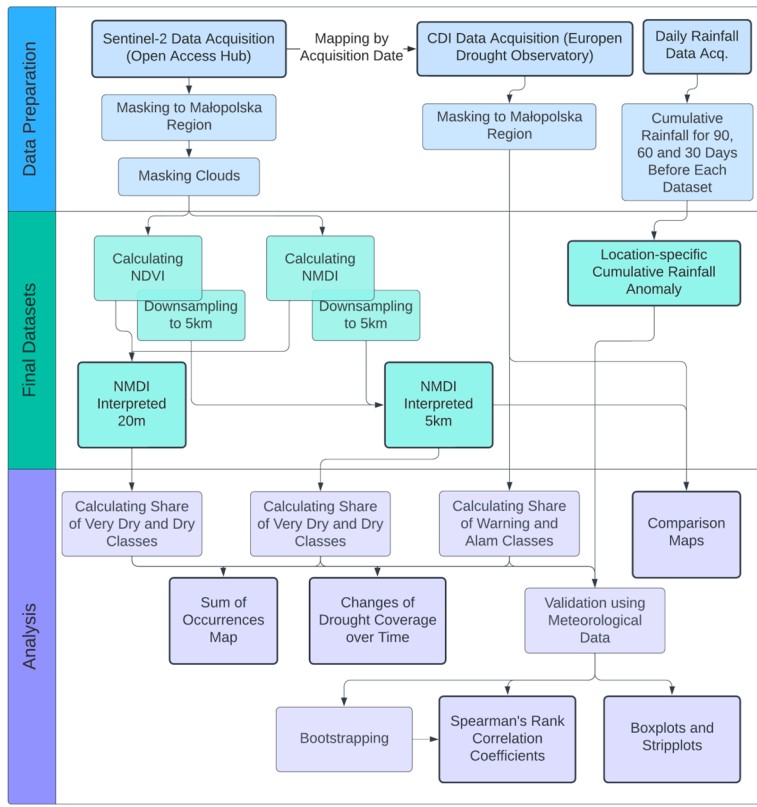

**Figure 3.** Diagram of the materials and methods used in this study.

## 2.1. Satellite Imagery

The research conducted within this study is based on level 2A imagery from Setninel-2 satellites. The mission consists of a constellation of two satellites, Sentinel-2A and Sentinel-2B, which acquire high spatial resolution (10–60 m) optical imagery [58]. The mission carries a wide-swath, high-resolution, multi-spectral imager (MSI) with 13 spectral bands, offering unprecedented perspectives on land and vegetation [59]. The primary objective of the Sentinel-2 mission is to provide high-resolution satellite data for land use monitoring, climate change and disaster monitoring, complementing other satellite missions, such as Landsat [60].

The period chosen for analysis encompasses the years 2018–2023. This selection was made due to the absence of level 2A data in the preceding years in the Open Access Hub, the platform operated by the European Space Agency for data provision. It was also challenging to find imagery with low cloud cover before 2018. In the years 2015–2017, there was only one satellite operating from Sentinel-2 constellation—Sentinel-2A.

Despite cloud and snow masking during data processing, appropriate imagery selection enhances the representativeness of results and statistics. Attention was also given to choosing images from similar periods within each year. Taking these considerations into account, four dates were selected for each year, and their detailed compilation is presented in the table below (Table 1). Almost all data (exceptions described below Table 1) were acquired by Sentinel-2 satellites under relative orbit number 036. Furthermore, measurements from both satellites were utilized.

**Table 1.** Dates for selected satellite imagery. S2A—imagery acquired by Sentinel-2A. S2B—imagery acquired by Sentinel-2B.

| Year | Date 1 | Date 2 | Date 3 | Date 4 |
|---|---|---|---|---|
| 2018 | 20 April S2A | 19 June S2A | 23 August S2B | 6 November S2A |
| 2019 | 31 March S2B | 9 June S2B | 28 August S2B | 27 October S2B |
| 2020 | 9 April S2A | 13 June S2B | 22 August S2B | 25 November S2A |
| 2021 | 9 April S2B | 18 June S2B | 6 September S2B | 31 October S2A |
| 2022 | 25 March S2B | 3 June S2B | 27 August S2A | 31 October S2B |
| 2023 | 23 April * | 3 June S2A | - | - |

* It was not possible to select images from a single day; therefore, images from 22 April and 24 April 2023 were used. The images from 22 April (S2B, orbit 079) had a sufficiently low level of cloud coverage; however, orbit number 079 did not cover the entire Małopolska region, so the gaps were filled with data from 24 April (S2A, orbit 036).

## 2.2. Normalized Multi-Band Drought Index and Selected Interpretation

The normalized multi-band drought index (NMDI) is used to assess moisture levels, addressing issues with the soil reflectance factor, which affect other vegetation-based indices (e.g., NDWI, NDVI) [61]. Wang and Qu [61] introduced the NMDI as a novel approach to monitor drought using satellite remote sensing. They aimed to provide a more accurate and robust measure, especially in areas with issues related to soil reflectance. NMDI addresses a critical challenge faced by other vegetation-based indices, namely the influence of soil reflectance. This can be especially problematic in areas with sparse vegetation cover, where soil has a significant impact on the reflectance values obtained by satellite sensors.

The formula for NMDI using Sentinel-2 spectral bands' notation is as follows:

$$NMDI = (B08 - (B11 - B12))/(B08 + (B11 - B12))$$

where B08 refers to the near-infrared band (VNIR, 842 nm), whereas B11 and B12 refer to the short-wave infrared bands (SWIR, 1610 nm and 2190 nm, respectively).

The interpretation created by L. Wang, J. Qu and X. Hao [62] was used in this study. It is based not only on NMDI but also utilizes the normalized difference vegetation index

(NDVI) [63] to differentiate areas with vegetation and bare soil and understand moisture content in the context of vegetation health.

The formula for NDVI using Sentinel-2 spectral bands' notation is as follows:

$$NDVI = (B08 - B04)/(B08 + B04)$$

where B08 refers to the near-infrared band (VNIR, 842 nm), B04 refers to the visible red band (665 nm), whereas B11 and B12 refer to the short-wave infrared bands (SWIR, 1610 nm and 2190 nm, respectively).

The classification divides pixels into 4 classes: Very Wet, Wet, Dry and Very Dry. The formula used for the interpretation of NMDI is described in Table 2.

**Table 2.** Formula used for the interpretation of NMDI [62].

| Class | Vegetation (NDVI $\geq$ 0.4) | Soil (NDVI < 0.4) |
| --- | --- | --- |
| Very Wet | >0.6 | <0.3 |
| Wet | 0.4–0.6 | 0.3–0.5 |
| Dry | <0.4 | >0.5 |
| Very Dry | <0.2 | 0.7–0.9 |

By incorporating NDVI, this interpretation provides context for the moisture levels not only of vegetation but also of soil. NMDI responds oppositely to soil moisture compared to vegetation water content. For instance, for vegetation, higher values mean higher water content, but for bare soil, higher values mean less soil moisture. Additionally, dry classes in an area with healthy vegetation (high NDVI) might be more concerning than the same type of class in an area with typically low vegetation or bare soil. This combined approach allows for better targeted interventions and can aid in identifying areas in need of immediate attention due to drought or other water-related concerns. The classification was performed on data with the original spatial resolution (20 m). Furthermore, a down-sampling (the weighted average of all non-NODATA contributing pixels) of NMDI values was carried out to obtain pixels with the same resolution as CDI (5 km).

*2.3. Combined Drought Indicator*

In this study, the third version [64] of the combined drought indicator (CDI) was employed. It has been implemented in the European Drought Observatory (EDO) and is created by combining three drought indicators:

- standardized precipitation index (SPI),
- soil moisture anomaly (SMA),
- fraction of absorbed photosynthetically active radiation anomaly (FAPAR anomaly).

The CDI is calculated at specific intervals—10 days—resulting in values determined for specific days: the 1st, 11th and 21st day of each month. The final product generated for this indicator is a raster with a spatial resolution of 5 km, where each individual pixel contains information about the class indicating the environmental condition in drought-affected areas. The individual classes have the following interpretations:

- 0—No Drought: normal conditions,
- 1—Monitoring: precipitation deficit,
- 2—Warning: negative effects affecting soil moisture, typically caused by precipitation deficit,
- 3—Alarm: negative effects impacting vegetation growth, usually due to precipitation deficit and reduced soil moisture,
- 4—Recovery: post-drought period, both meteorological conditions and vegetation growth return to normal,
- 5—Temporary Recovery of Soil Moisture: soil moisture conditions are above the drought threshold but still insufficient to consider the drought episode conclusive,

- 6—Temporary Recovery of Vegetation Condition: the vegetation condition is above the drought threshold but still insufficient to consider the episode conclusive.

### 2.4. Methods Used for Comparison between NMDI and CDI

Both the combined drought indicator (CDI) and the interpreted normalized multi-band drought index (NMDI) are presented in the form of classified (discrete) values for individual pixels. To compare the extents of the Małopolska region, which may experience drought during a specific period between both indices, it was decided to assess the proportion of the following classes in the total number of pixels covering the Małopolska region:

- For CDI—share of Warning and Alarm classes,
- For NMDI—share of Dry and Very Dry classes.

The data in this form were then compared on a line chart, where their variability over time is visible, allowing for a visual comparison of their fluctuations. These data were also examined in tabular form, enabling the identification of periods where the indices indicated the most similar environmental conditions.

A spatial analysis was conducted using CDI and NMDI maps for selected dates. Additionally, for each pixel, counts of the relevant classes were also presented; for CDI, these were the Warning and Alarm classes, and for NMDI, they were the Dry and Very Dry classes. This allowed for the definition of areas where these classes occur most frequently, and consequently, the presentation of regions, which are most susceptible to drought (according to the utilized index).

The table below presents a compilation of satellite imagery acquisition dates with the dates of CDI, which were used for comparison (Table 3).

**Table 3.** Compilation of acquisition dates for satellite imagery and CDI used for comparison.

| Year | Type | Date 1 | Date 2 | Date 3 | Date 4 |
|------|------|--------|--------|--------|--------|
| 2018 | Sat. img. | 20 April | 19 June | 23 August | 6 November |
|      | CDI | 21 April | 21 June | 21 August | 11 November |
| 2019 | Sat. img. | 31 March | 9 June | 28 August | 27 October |
|      | CDI | 1 April | 11 June | 1 September | 1 November |
| 2020 | Sat. img. | 9 April | 13 June | 22 August | 25 November |
|      | CDI | 11 April | 11 June | 21 August | 21 November |
| 2021 | Sat. img. | 9 April | 18 June | 6 September | 31 October |
|      | CDI | 11 April | 21 June | 11 September | 1 November |
| 2022 | Sat. img. | 25 March | 3 June | 27 August | 31 October |
|      | CDI | 21 March | 1 June | 1 September | 1 November |
| 2023 | Sat. img. | 23 April | 3 June | - | - |
|      | CDI | 21 April | 1 June | - | - |

### 2.5. Methods Used for Comparison with Meteorological Data

The verification of the results utilizes in situ meteorological measurements from the open data from the Polish Institute of Meteorology and Water Management (IMGW). All calculations are based on daily rainfall from 97 stations located throughout Małopolska (Figure 4).

CSV files were downloaded from the archive and then processed into the appropriate format. A preliminary dataset was established as a table, where, for each station and each day, the total precipitation was presented. The cumulative rainfall for 30, 60 and 90 days before each date, from which the utilized indices originated, was then calculated based on this dataset and converted to rainfall anomaly to account for the differences in characteristics of specific locations. The rainfall anomaly was calculated by subtracting a mean value of cumulative rainfall from a certain period (30, 60 or 90 days) registered at a specific station from all cumulative rainfall values from that period registered at

that station. Data were presented as box plots and strip plots to aid visual analysis, and Spearman's rank correlation coefficients were calculated based on the entire dataset and bootstrapped samples.

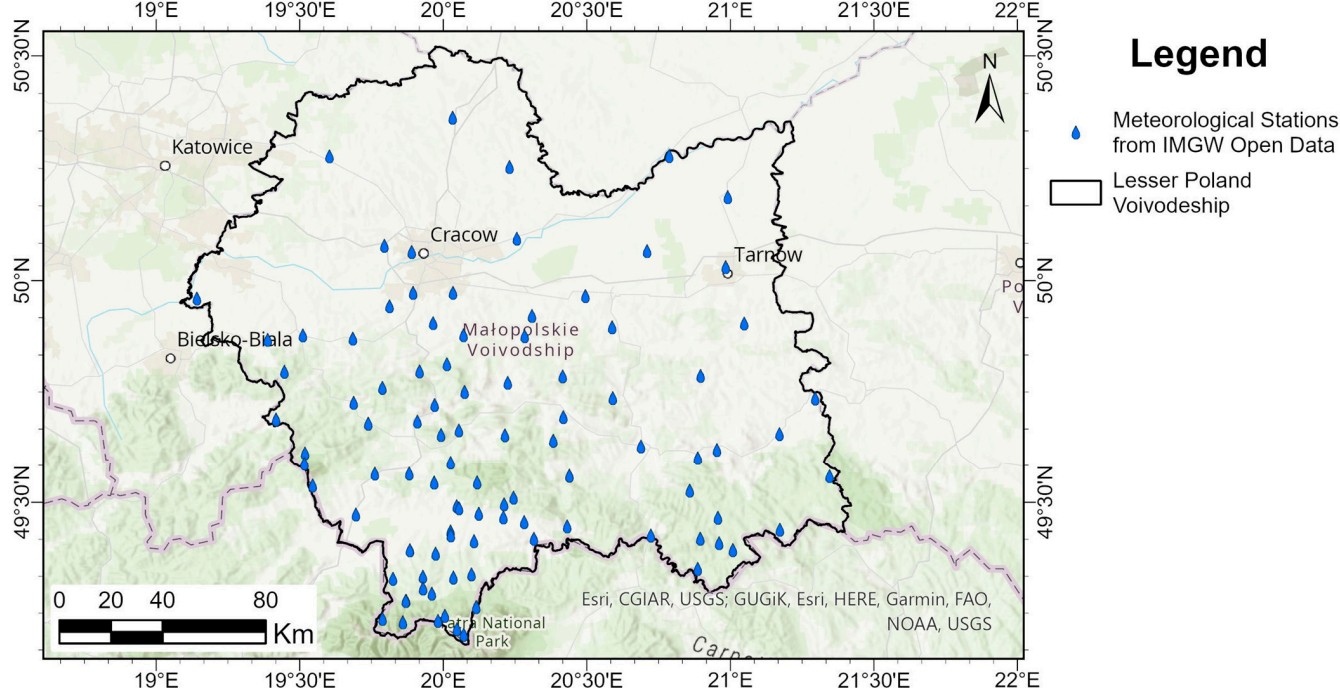

**Figure 4.** Location of meteorological stations in Małopolska (based on IMGW open data).

### 2.6. Data Processing Environment

Data acquisition, processing and visualization in this study were performed using the Python programming language. The main libraries utilized included sentinelsat (version 1.1.1), rasterio (version 1.3.3), numpy (version 1.23.5), scipy (version 1.10.1), geopandas (version 0.12.1), matplotlib (version 3.6.2) and seaborn (version 0.13.2).

## 3. Results

### 3.1. Comparison between NMDI and CDI

In terms of analysis of the level of coverage of the Małopolska region by classes indicating drought occurrence, the compared indices yield extremely divergent results. The variability in these levels over time is entirely unrelated, similarly to the case of observed peaks (Figure 5). The fluctuations observed for CDI—Warning and Alarm increase their range only from the year 2020 onwards (this is particularly evident in their lower limit). Regarding NMDI, it is challenging to discern any trends over the analyzed years, with the only anomaly being a peak in the data from 25 March 2022, which is distinctly noticeable. Interestingly, for CDI, the values did not deviate beyond the bounds of their standard fluctuations during this time. Instead, a peak emerges for coverage in the Alarm class in CDI, but only in subsequent observed dates.

The indices compared provide such disparate insights that it is difficult to apply them to a general assessment of the environment in terms of the possible drought phenomenon in the Małopolska region. The fluctuations occurring for both indices can only indicate that the conditions prevailing in this area did not change over the years analyzed.

Even when examining data from periods showing the highest similarity between indices (Table 4), it is impossible to pinpoint areas, which may be more susceptible to drought occurrence (Supplementary File S1). Moreover, for the first eight periods with the highest similarity, the level indicated by the indices is not high. Only the ninth period—25 March 2022—shows higher coverage by classes indicative of drought. However, upon scrutinizing the spatial distribution of individual classes during this period, it is impossible

to define areas with an elevated risk of drought consistently for both indices. Even assuming that there may be a delayed response in the case of CDI—the maps were compared with a lag of one period (Figure 6)—the areas with the highest class for NMDI and CDI are in completely different locations.

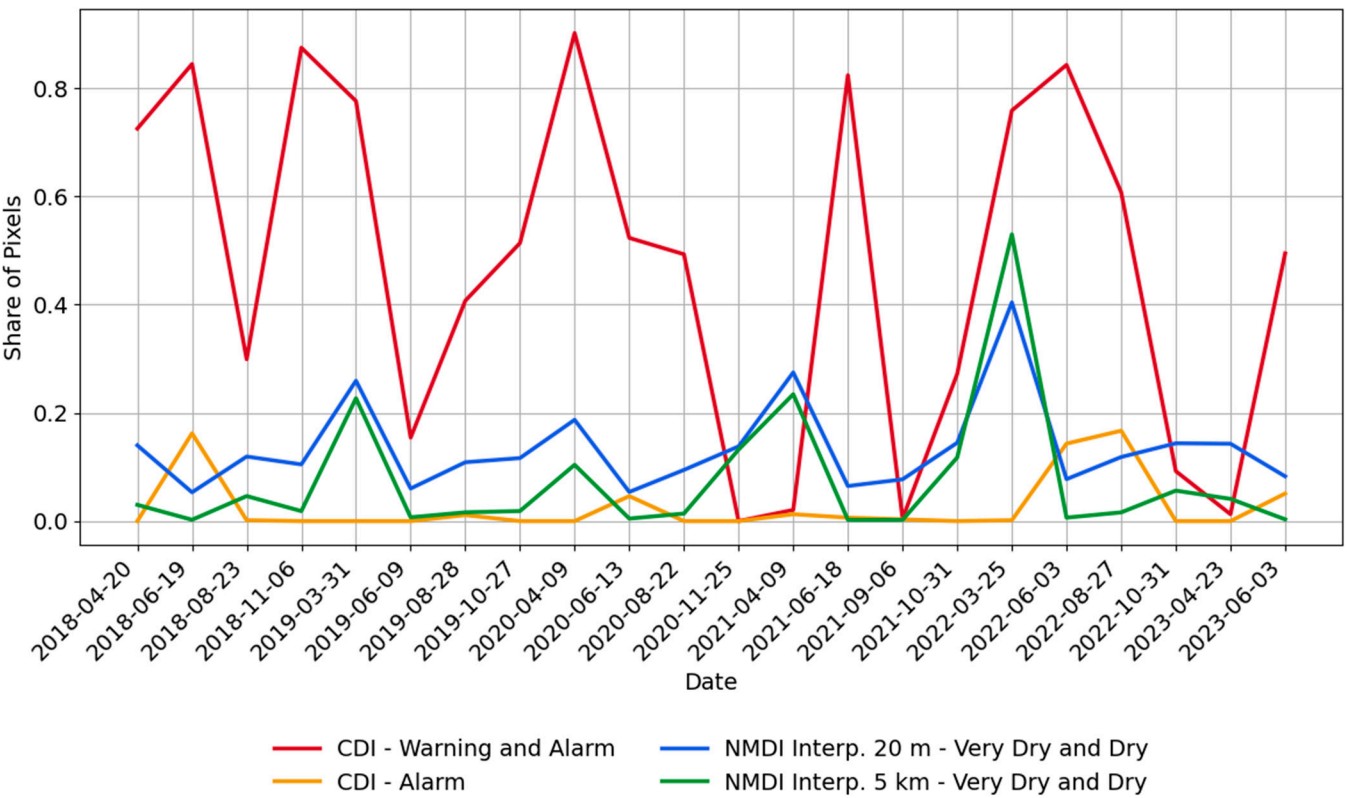

**Figure 5.** Coverage of the Małopolska region with areas with an elevated risk of drought indicated by NMDI and CDI. The share of selected classes in the total number of pixels for the Małopolska region for CDI and interpreted, down-sampled NMDI (CDI—Warning and Alarm classes; NMDI Interpreted—Dry and Very Dry classes).

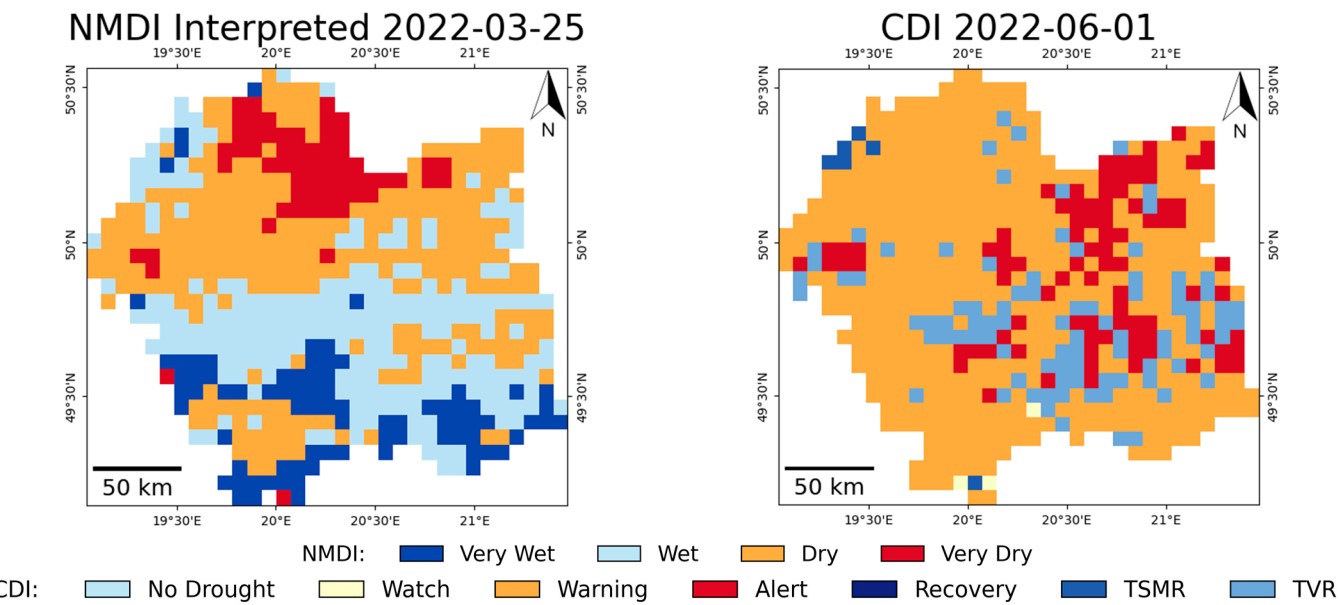

**Figure 6.** Comparative maps for interpreted, down-sampled NMDI for 25 March 2022 and CDI for 1 June 2022 (TSMR—Temporary Soil Moisture Recovery, TVR—Temporary Vegetation Recovery).

**Table 4.** Coverage of the Małopolska region with areas with an elevated risk of drought indicated by NMDI and CDI. The share of selected classes in the total number of pixels for the Małopolska region for CDI, interpreted NMDI (20 m pixel) and interpreted, down-sampled NMDI (5 km pixel) (CDI—Warning and Alarm classes; NMDI Interpreted—Dry and Very Dry classes). Values sorted from smallest to largest according to the Difference column values.

| Date | CDI 5 km | NMDI 5 km | NMDI 20 m | Difference * |
|---|---|---|---|---|
| 6 September 2021 | 0.00 | 0.08 | 0.00 | 0.04 |
| 31 October 2022 | 0.09 | 0.14 | 0.06 | 0.04 |
| 23 April 2023 | 0.01 | 0.14 | 0.04 | 0.08 |
| 9 June 2019 | 0.15 | 0.06 | 0.01 | 0.12 |
| 25 November 2020 | 0.00 | 0.14 | 0.13 | 0.14 |
| 31 October 2021 | 0.27 | 0.14 | 0.12 | 0.14 |
| 23 August 2018 | 0.30 | 0.12 | 0.05 | 0.22 |
| 9 April 2021 | 0.02 | 0.27 | 0.23 | 0.23 |
| 25 March 2022 | 0.76 | 0.40 | 0.53 | 0.29 |
| 28 August 2019 | 0.41 | 0.11 | 0.02 | 0.34 |
| 22 August 2020 | 0.49 | 0.09 | 0.01 | 0.44 |
| 27 October 2019 | 0.51 | 0.12 | 0.02 | 0.45 |
| 3 June 2023 | 0.49 | 0.08 | 0.00 | 0.45 |
| 13 June 2020 | 0.52 | 0.05 | 0.00 | 0.49 |
| 31 March 2019 | 0.78 | 0.26 | 0.23 | 0.53 |
| 27 August 2022 | 0.61 | 0.12 | 0.02 | 0.54 |
| 20 April 2018 | 0.72 | 0.14 | 0.03 | 0.64 |
| 9 April 2020 | 0.90 | 0.19 | 0.10 | 0.76 |
| 18 June 2021 | 0.82 | 0.06 | 0.00 | 0.79 |
| 3 June 2022 | 0.84 | 0.08 | 0.01 | 0.80 |
| 6 November 2018 | 0.87 | 0.10 | 0.02 | 0.81 |
| 19 June 2018 | 0.84 | 0.05 | 0.00 | 0.82 |

* The difference between CDI and NMDI is calculated as the average of the absolute values for two differences—between CDI and NMDI 5 km and between CDI and NMDI 20 m.

Concerning the sum of occurrences of Dry and Very Dry classes for NMDI in various locations, higher values are evident in the northern part of the region (Figure 7). This implies that in those areas, classes indicative of drought occurrence appeared most frequently during the analyzed period. Individual pixels with higher values in other locations may result from the presence of larger water reservoirs or exposed rocks (mountainous regions without vegetation are located in the southern part of the region). For CDI, in the sum of occurrences of Warning and Alarm classes, the area with the highest values is located in the central part of Małopolska (Figure 8). Elevated values in the north, present in the sum for NMDI, are in no way reflected in the case of CDI. Regarding counts of only the Alarm class, elevated values are visible in the southeastern part of Małopolska. Some of them overlap with observations for NMDI. However, due to the displacement, they are not the result of the same changes occurring in the natural environment.

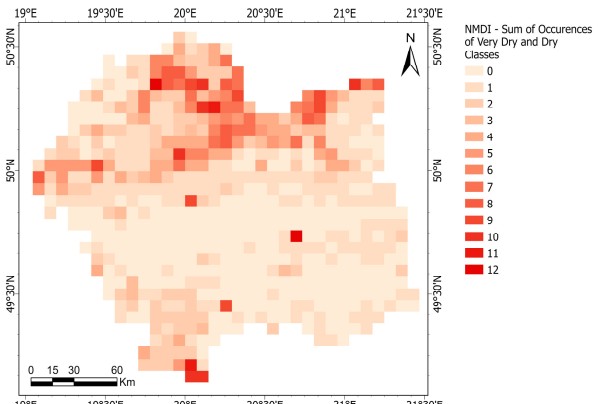

**Figure 7.** Sum of occurrences of Dry and Very Dry classes for interpreted NMDI.

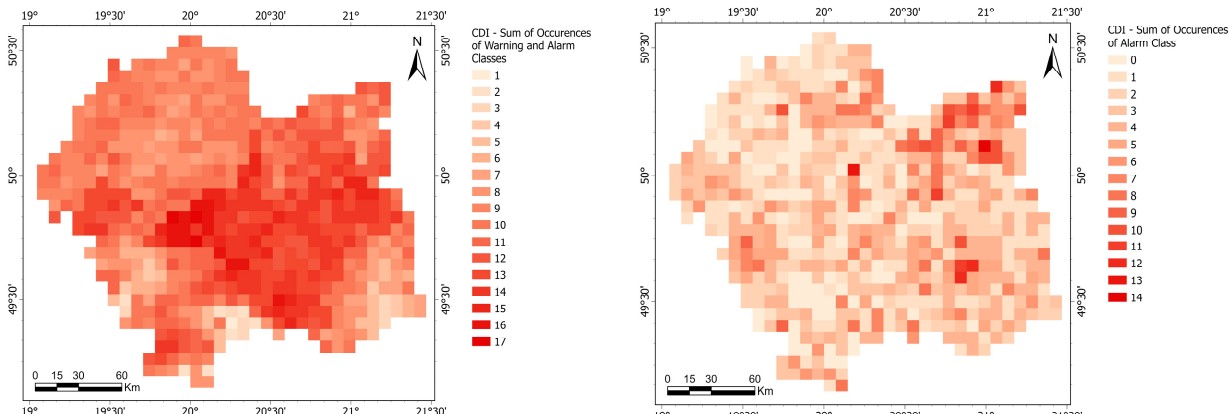

**Figure 8.** (**left**), sum of occurrences of Warning and Alarm classes for CDI; (**right**), sum of occurrences of Alarm class only.

Despite analyzing other periods as well (all available in Supplementary File S1), no dependencies were found between these indices. They did not exhibit similarities either in temporal variability for the share of dry regions or in their locations. With the analyzed data, it was also not possible to identify regions where dry conditions occur more frequently consistently for both indices.

On the other hand, it appears that, in the case of NMDI, elevated values of the sum of occurrences in the northern part of the region may be related to land use. The northern part of the province is more intensively utilized for agriculture (Figure 2). However, this cannot explain areas with elevated sums for CDI. The central part of the region has rather mixed land use. Interestingly, heavily urbanized areas do not stand out in the indices; there are no conspicuous values for either of them.

Due to limitations in the frequency of CDI datasets, there is a certain gap between the acquisition of satellite imagery used to calculate NMDI and the date for which CDI is calculated. Utilizing a scatter plot (Figure 9), no relationship was observed between differences in the share of classes considered dry and the number of days, which differed between satellite data acquisition for NMDI and the calculation of CDI. However, it should be noted that the number of data points is insufficient for conducting a proper statistical analysis to exclude such dependency.

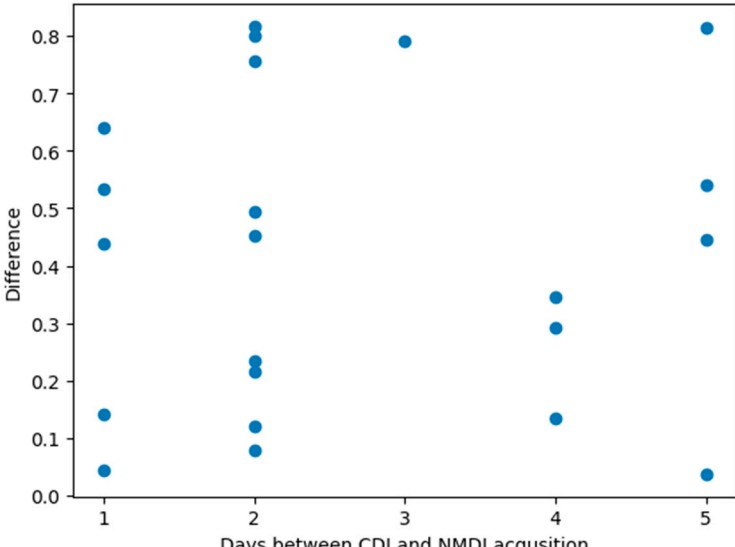

**Figure 9.** Scatter plot for the number of days between acquisition of satellite imagery for NMDI and calculation of CDI and differences in the share of drought areas between CDI and NMDI presented in Table 4.

### 3.2. Comparison with Meteorological Data

In order to gain additional information regarding the performance of both indices, the CDI and NMDI classes from pixels containing meteorological stations (total of 97 locations) were compared with one of the conventional drought predictors: rainfall anomaly from 90 days prior to each dataset's acquisition.

A visual inspection of the data (Figure 10) shows negative correlation between rainfall anomaly and NMDI and a complicated non-linear and non-monotonic relationship between rainfall anomaly and CDI. The inspection also shows that the distribution of observations between the classes is extremely uneven. For each index, the vast majority of observations fall into two classes: Very Wet and Wet in the case of NMDI and No Drought and Warning in the case of CDI.

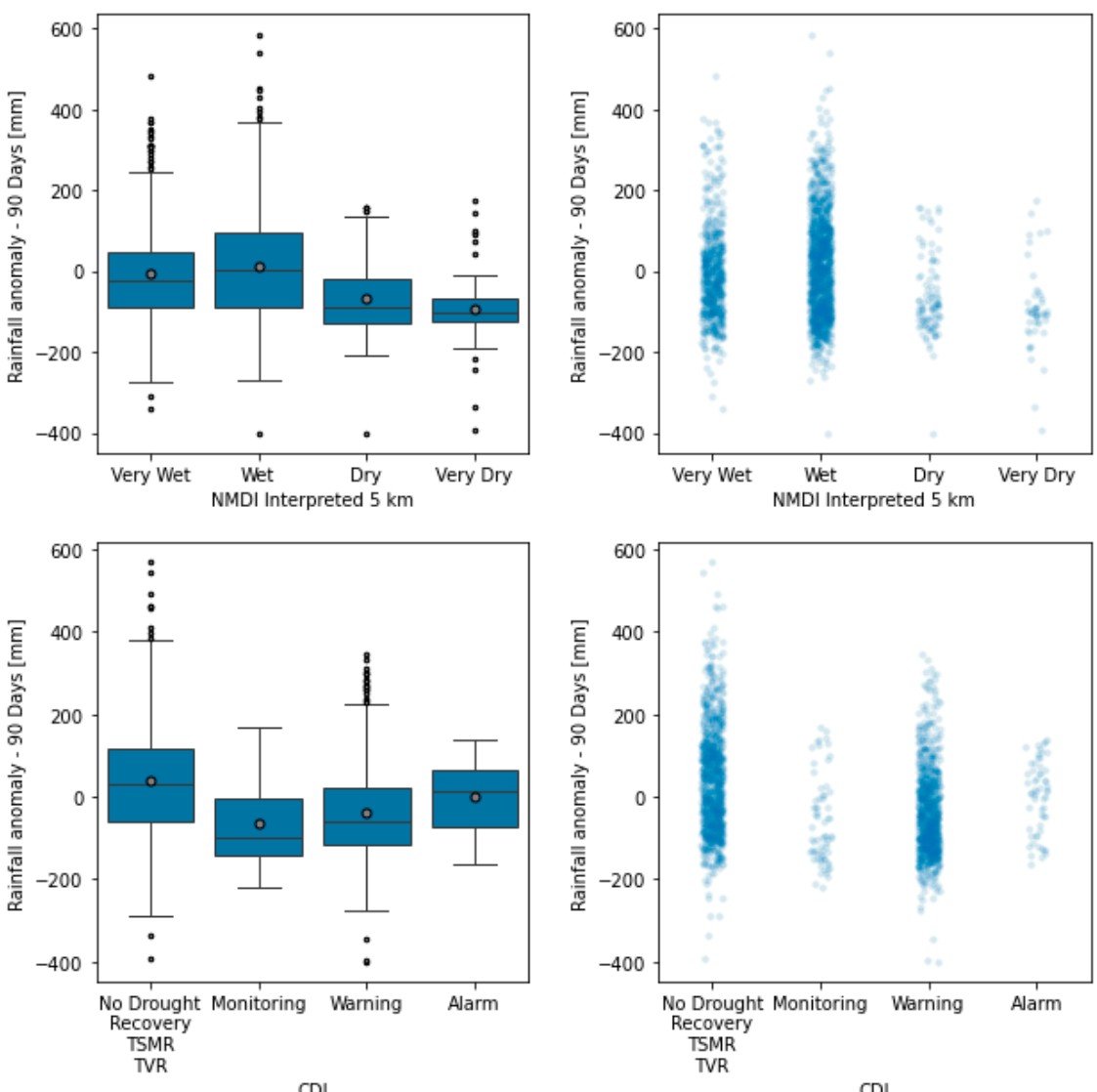

**Figure 10.** Comparison of NMDI and CDI classifications with rainfall anomaly from 90 days prior to each dataset's acquisition; TSMR—Temporary Soil Moisture Recovery; TVR—Temporary Vegetation Recover.

Due to extremely uneven data distribution between individual classes, the observed relationships are not reflected in the calculated correlation coefficients. The values of Spearman's rank correlation coefficient are equal to −0.0363 for NMDI and −0.2965 for CDI, which indicates a much higher negative correlation between rainfall anomaly and CDI than NMDI and contradicts the relationships observed in the plots.

In order to address this problem, a bootstrap method with disproportionate stratified random sampling was used. A total of 5000 samples were drawn separately for the NMDI and CDI datasets, ensuring that each class contributed 50 observations for each sample selected with replacement (200 observations in total). Based on each sample, Spearman's rank correlation coefficient was calculated. The results (Figure 11) indicate a much higher negative correlation between rainfall anomaly and NMDI than CDI. The 95% confidence intervals of Spearman's rank correlation coefficient were $[-0.4491, -0.1992]$ for NMDI and $[-0.1731, 0.0792]$ for CDI, with mean values of $-0.3256$ for NMDI and $-0.0477$ for CDI.

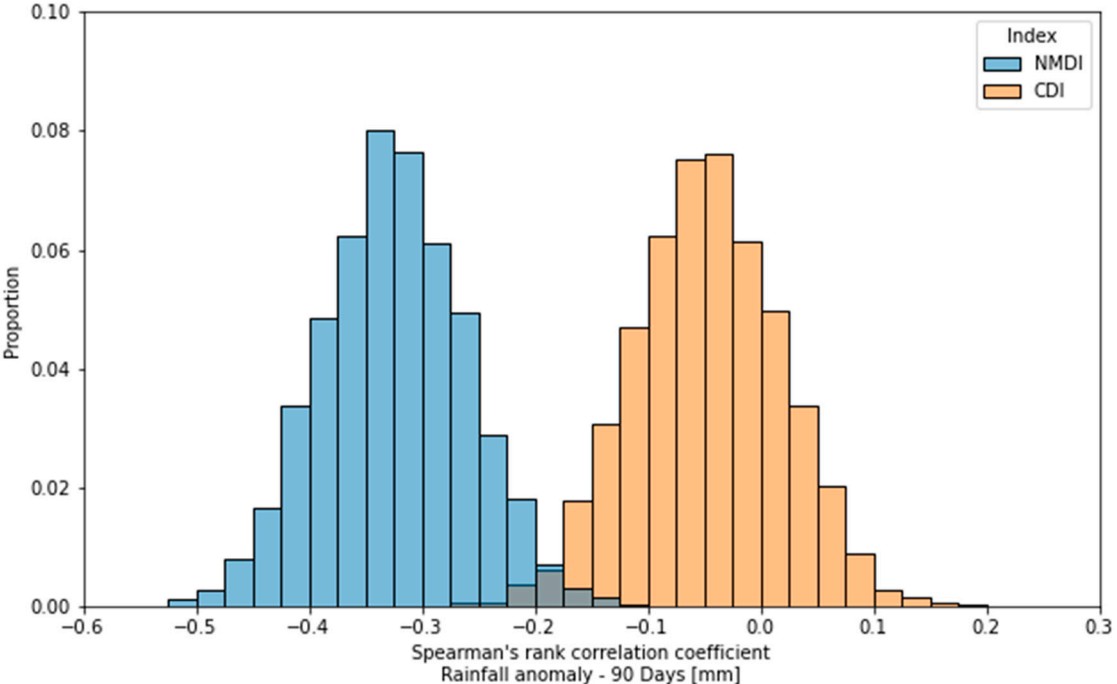

**Figure 11.** Distribution of Spearman's rank correlation coefficients obtained from bootstrapping.

A similar analysis was performed for rainfall anomalies calculated for 30 and 60 days prior to each dataset's acquisition (Supplementary File S2). In both cases, the results were similar to those presented above.

## 4. Discussion

The study of drought has been the subject of extensive research, with a focus on utilizing remote sensing and drought indices to monitor and understand drought events. Aghakouchak [65] discussed the progress, challenges and opportunities encountered in remote sensing of drought, highlighting the limitations of satellite missions and sensors in providing sufficient data for studying droughts from a climate perspective. This article also points out that remote sensing has numerous limitations, mainly due to the time resolution of the data. Blauhut [66] emphasized the wide-ranging transboundary, environmental and socio-economic impacts of drought in Europe, particularly on sectors such as agriculture, energy production, public water supply and water quality. Furthermore, Stagge [67] observed an increasing divergence in the observed drought indices across Europe, with Southern Europe experiencing increasing drought frequencies and Northern Europe experiencing decreasing frequencies. Nyayapathi [68] emphasized the robustness and effectiveness of remote sensing-based studies in monitoring and mapping droughts compared to conventional ground survey methods, indicating the potential of remote sensing in drought assessment. The observation of this article is a confirmation that remote sensing is an important source of knowledge regarding drought due to the ability to observe large areas without the need to perform in situ measurements. These studies collectively

demonstrate the significance of remote sensing and drought indices in understanding the complex and diverse patterns of drought across Europe.

All the data employed in this manuscript possess unique characteristics, rendering their comparative analysis challenging; nonetheless, employing multi-index analyses may yield more insightful information. For the calculated NMDI index, which is based on optical data, the challenge lies in selecting imagery to minimize the impact of clouds. Clouds significantly influence the results of analysis, and their masking is a crucial element of data processing [69]. In this research, the criteria for selecting images with minimal cloud coverage resulted in delineation of only four overlapping periods for each year. However, it is important to acknowledge that the criteria employed were exceedingly stringent. By lowering the threshold, the quantity of satellite imagery available could markedly rise; however, this would entail excluding a more substantial portion of the area, thereby diminishing the representativeness of the findings.

In Ref. [70], Sentinel-2's superior spatial, spectral and temporal resolution is highlighted as a pivotal asset in drought assessment, offering enhanced capabilities over contemporaneous platforms, such as Landsat-8. The distinct advantage of Sentinel-2 is attributed to its red-edge band, which provides high precision in detecting vegetation responses to drought, surpassing conventional indices, such as NDVI. Furthermore, the potential application of advanced machine-learning and deep-learning algorithms in conjunction with Sentinel-2 data could further enhance the accuracy of classifying surface water, wetlands, vegetation and general land cover, thereby strengthening the capabilities of drought monitoring and management through the use of NMDI. This aligns with and expands upon the findings of our study, underscoring the integral role of Sentinel-2 in comprehensive drought analysis.

The amount of data used in this article for each year may be insufficient, which could possibly lead to omission of some variability in the studied phenomena. In the case of CDI, new data appear three times a month, potentially providing a better picture of the changes occurring in the environment. However, a spatial resolution of 5 km does not allow for a detailed analysis of these phenomena. Furthermore, some of the source datasets on which CDI is based have an even lower spatial resolution. The precipitation data used have a pixel size of 0.25 degrees [25]. High-resolution images are a valuable source of information, but usually, either high spatial or temporal resolution is available, never both [71]. NMDI, due to its superior spatial resolution, possesses a notable advantage. The distinct differences between NMDI and CDI carry critical implications for practical applications in drought management. NMDI's fine spatial resolution (20 m) allows for detailed, localized drought analysis, enabling precise management interventions in small, critically impacted areas. Meanwhile, CDI, with a spatial resolution of about 5 km, offers a broad overview, beneficial for larger scale drought planning and management, albeit potentially missing localized anomalies. Ref. [72] provides empirical support for the findings presented in this research, particularly in the context of using optical data for drought monitoring—in this case, in agricultural settings. It underscores the methodological strengths and practical implications of NMDI, validating the current research's approach in leveraging this index for detailed and accurate assessment of drought impacts. As an example, for limited use of low spatial resolution imagery, soil moisture products based on microwaves currently lack sufficient spatial resolution to be useful for applications in irrigation management or flood predictions [73]. West et al. highlight spatial resolution as one of the factors identified for future development of drought assessment [22].

CDI also utilizes historical data, so the information obtained refers to a wider time range. On the other hand, it treats each pixel separately, making it impossible to infer from it that drought is at a higher level in one area than in another because these values are always referenced individually to the historical conditions prevailing there. In contrast, NMDI allows us to draw such conclusions, since the methodology focuses on data from a specific time, making it universal and absolute.

In examining the comparisons of the indices, stark contrasts were identified, notably in the spatial distribution of regions with higher index values, which indicate potential drought occurrences (Supplementary File S1), and in the overall area defined as potentially drought-prone (expressed as a percentage of pixels; Figure 5).

NMDI, which utilizes multi-spectral imagery, facilitates the measurement of land surface absorption levels across various electromagnetic wave ranges, thereby enabling the assessment of factors such as vegetation chlorophyll content and water absorption, contingent upon the spectral band. Conversely, CDI employs a distinct dataset comprising the standardized precipitation index (SPI), soil moisture anomaly (SMA) and fraction of absorbed photosynthetically active radiation anomaly (FAPARA).

A pivotal distinction exists in the reference periods employed by the indices. NMDI does not utilize reference data, with its values being absolute and its maximum analysis period spanning from 2015 to 2023, employing Sentinel-2 datasets. In contrast, CDI draws upon three relative information sources, using reference periods to pinpoint anomalies. SPI employs a reference period 1981–2010; SMA uses a reference period 1996–2018; and FAPARA utilizes a reference period 2001–2018 [74]. This endows the indices with markedly different perspectives in delineating drought-affected areas. While NMDI permits the observation of changes solely within the analyzed period, CDI yields results, which portray deviations in the current conditions relative to the reference periods. If the chosen reference periods are inappropriately wide or encompass anomalous periods, the resultant data may neglect significant environmental changes.

Furthermore, the choice of data inputs and reference periods varies significantly between the two. NMDI provides an absolute, real-time snapshot of the current conditions, imperative for immediate response scenarios. Contrastingly, CDI utilizes historical reference periods, offering a normalized perspective, which aids in understanding the current drought conditions relative to historical norms.

## 5. Conclusions

The meticulous comparison between the normalized multi-band drought index (NMDI) and the combined drought indicator (CDI) performed within this study unveiled significant insights into the drought dynamics of the Małopolska region. The findings illuminate the distinct advantages and synergies of utilizing both indices for a more nuanced understanding of drought severity. The research demonstrated a robust correlation between satellite-derived indices and in situ observations, underscoring the potential of an integrated approach in enhancing drought detection and monitoring capabilities. Notably, the study revealed that the indices depict different dynamics of drought levels, as well as the location of regions more prone to its occurrence. The convergence of these indices in reflecting true drought conditions suggests a compelling avenue for improving the precision of drought assessments. As a result, the study advocates for the integration of NMDI and CDI into existing drought management frameworks to support more informed decision making. Future research is encouraged to focus on refining these indices and exploring their application in other regions, aiming to bolster global drought resilience through advanced remote sensing technologies.

**Supplementary Materials:** The following supporting information can be downloaded at: https://www.mdpi.com/article/10.3390/rs16050836/s1, Supplementary File S1: The full set of CDI and NMDI comparison maps for 22 different dates, which were analyzed in this study. Supplementary File S2: A comparison of NMDI and CDI with rainfall anomaly computed based on cumulative rainfall from 30, 60 and 90 days prior to each dataset acquisition.

**Author Contributions:** Conceptualization, M.L. and J.S.; methodology, M.L. and J.S.; software, J.S.; validation, M.W., M.L. and J.S.; formal analysis, J.S. and M.L.; investigation, M.L., J.S., K.A., M.W., A.M.-O. and A.O.; data curation, J.S.; writing—original draft preparation, J.S., M.L. and K.A.; writing—review and editing, A.M.-O. and A.O.; visualization, J.S. and M.W.; supervision, M.L. All authors have read and agreed to the published version of the manuscript.

**Funding:** The work was financed within the framework of statutory research of the Faculty of Geology, Geophysics and Environmental Protection, AGH University of Krakow.

**Data Availability Statement:** Original data used in this research are available in a publicly accessible repository (edo.jrc.ec.europa.eu/ for CDI; scihub.copernicus.eu/ for Sentinel-2).

**Conflicts of Interest:** The authors declare no conflicts of interest.

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
