# Peer review of "Spatial Insights into Drought Severity: Multi-Index Assessment in Małopolska, Poland, via Satellite Observations"

_remotesensing, doi:10.3390/rs16050836_

Round 1

Reviewer 1 Report

Comments and Suggestions for Authors

The article is well-written and presents an interesting study on the use of high spatial resolution multispectral imagery for drought assessment in Lesser Poland. The authors compare the Normalized Multi-Band Drought Index (NMDI) with the Combined Drought Indicator (CDI) and analyze their temporal and spatial variability and correlation with meteorological data. But, the article could be improved by addressing the following issues:

1. The authors should clarify the motivation and novelty of their study, as the use of NMDI and CDI for drought monitoring has been previously reported in the literature. What are the specific research questions and hypotheses that the authors aim to answer? How does their study contribute to the advancement of knowledge and practice in the field of drought assessment?

2. The authors should provide more details on the methodology and data used in their study. For example, how were the NMDI and CDI calculated and interpreted? What are the advantages and limitations of using Sentinel-2 imagery for NMDI? How were the meteorological data obtained and processed? How were the spatial and temporal comparisons and correlations performed? What are the sources of uncertainty and error in the analysis?

3. The authors should present and discuss the results more clearly and comprehensively. For example, what are the main findings and implications of the temporal and spatial variability and correlation of NMDI and CDI? How do they compare with previous studies and expectations? What are the possible explanations and mechanisms for the observed differences and similarities between the two indices? How do the results relate to the research questions and hypotheses?

4.  You should better refer to more references especially at the discussion part.

Author Response

Dear Reviewer,

we want to convey our profound gratitude for the thoughtful and detailed feedback you provided on our manuscript. Your expertise and keen insights have been pivotal in guiding significant improvements to our work.

We truly value the time and effort you've invested in reviewing our paper. Your constructive critiques and suggestions have not only illuminated areas for enhancement but have also enriched our understanding and approach to our research topic.

In response to your feedback, we have made comprehensive revisions to our manuscript. We believe these changes have greatly fortified our work, thanks to your invaluable contributions. Your dedication to fostering high-quality research is deeply appreciated.

Thank you once again for your essential role in refining our paper and for your commitment to the advancement of knowledge within our field.

Reviewer 2 Report

Comments and Suggestions for Authors

The authors compared 20m NMDI with 5km CDI in in Lesser Poland. The paper lacks of useful results and would be not interesting to readers. I recommend the article for rejection.

The specific comments:

1、It is suggested that the authors rewrite the abstract and introduction sections.  

2、Cumulative rainfalls from 90 days prior to each dataset's acquisition were computed (for both CDI and NMDI).I suggest the authors calculate rainfall anomaly rather than rainfall. Besides 90 days, the authors are suggested to add 30 days, 60 days and other cumulative period.

3、The authors compared the NMDI with CDI from figure 5 to figure 11.It is suggested that the authors calculate the NMDI anomaly rather than NMDI based on the sentinel-2 satellite images and then compared with CDI.  

Comments on the Quality of English Language

The English of MS requires extensive modification. 

Author Response

Dear Reviewer,

we want to convey our profound gratitude for the thoughtful and detailed feedback you provided on our manuscript. Your expertise and keen insights have been pivotal in guiding significant improvements to our work. Please refer to the attachment for the details and our answers.

We truly value the time and effort you've invested in reviewing our paper. Your constructive critiques and suggestions have not only illuminated areas for enhancement but have also enriched our understanding and approach to our research topic.

In response to your feedback, we have made comprehensive revisions to our manuscript. We believe these changes have greatly fortified our work, thanks to your invaluable contributions. Your dedication to fostering high-quality research is deeply appreciated.

Thank you once again for your essential role in refining our paper and for your commitment to the advancement of knowledge within our field.

Reviewer 3 Report

Comments and Suggestions for Authors

The paper investigated the use of two different indices (NMDI and CDI) in estimating drought severity in Lesser Poland. Overall, it's an interesting work, which may be published under a couple of highlights – corrections:

Abstract

Line 17: The abstract does not explicitly mention which satellite is employed for calculating the drought index.

The abstract is still insufficient; it does not provide information to convince the reader that the research is relevant.

Introduction

Line 41: Define acronym when first used

I suggest improving the introduction by providing a more detailed explanation of the study area ‘Lesser Poland’ and the specific challenges it encounters with regards to drought.

Materials and methods

Line 129: ‘Four dates…’ Justify this choice.

Figure 2 requires improvement to provide a clearer representation of topography, hydrological elements, land cover types and protected areas.

Line 132: ‘both satellites’ not clear, which ones?

Line 143: ’Wang and Qu’, provide a reference.

Ensure that all equations are numbered and referenced in the text.

Lines 152-163: The method of incorporating NDVI and determining the classification is not clearly explained. Additionally, the downsampling technique or interpolation method used is not explicitly detailed in the text.

Line 201: Please specify the number of weather stations.

Results

Overall, the approach looks like describes the results obtained. Readers would lose interest after spending time on a few pages. Thus, critical analysis is required.

Discussion

The discussion needs to be more compelling.

In terms of practical applications, what are the implications of the findings? What are the weaknesses of this research? What are the necessary future studies?

Author Response

Dear Reviewer,

we want to convey our profound gratitude for the thoughtful and detailed feedback you provided on our manuscript. Your expertise and keen insights have been pivotal in guiding significant improvements to our work. We redesigned and rewrote many elements of our manuscript. Please refer to the attachment for the details and our answers.

We truly value the time and effort you've invested in reviewing our paper. Your constructive critiques and suggestions have not only illuminated areas for enhancement but have also enriched our understanding and approach to our research topic.

In response to your feedback, we have made comprehensive revisions to our manuscript. We believe these changes have greatly fortified our work, thanks to your invaluable contributions. Your dedication to fostering high-quality research is deeply appreciated.

Thank you once again for your essential role in refining our paper and for your commitment to the advancement of knowledge within our field.

Reviewer 4 Report

Comments and Suggestions for Authors

This manuscript has certain research significance, but there is no new idea or method. There are some problems in the manuscript that need to be further answered or improved, as follows:

1. Adding the band information of Sentinel-2 is suggested.

2. NMDI for each period was calculated using only one scene image. It remains to be seen whether it will be affected by factors such as cloud cover.

3. It is recommended to add more information about CDI, including the timescale it represents. The spatial resolution of CDI is 5km. It is suggested to use meteorological station data to achieve higher spatial resolution results.

4. As can be seen from the manuscript, CDI may represent a cumulative result over a period of time, or a standardized anomaly. But NMDI calculations represent instantaneous information, can the two results be compared?

Author Response

(The authors gave the same response as above.)

Round 2

Reviewer 2 Report

Comments and Suggestions for Authors

I suggest acceptance in present form.

Reviewer 3 Report

Comments and Suggestions for Authors

Thanks for your revisions, I have no additional comments.

Reviewer 4 Report

Comments and Suggestions for Authors

I suggest accept it to publish in the journal.